# Enhanced Convolutional Neural Network for In Situ AUV Thruster Health Monitoring Using Acoustic Signals

**DOI:** 10.3390/s22187073

**Published:** 2022-09-19

**Authors:** Sang-Jae Yeo, Woen-Sug Choi, Suk-Yoon Hong, Jee-Hun Song

**Affiliations:** 1Department of Naval Architecture and Ocean Engineering, Seoul National University, Seoul 08826, Korea; 2Institute of Engineering Research, Seoul National University, Seoul 08826, Korea; 3Department of Ocean Engineering, Korea Maritime and Ocean University, Busan 49112, Korea; 4Department of Naval Architecture and Ocean Engineering, Chonnam National University, Yeosu 59626, Korea

**Keywords:** autonomous underwater vehicle (AUV), convolutional neural network (CNN), thruster health monitoring, wavelet transform

## Abstract

As the demand for ocean exploration increases, studies are being actively conducted on autonomous underwater vehicles (AUVs) that can efficiently perform various missions. To successfully perform long-term, wide-ranging missions, it is necessary to apply fault diagnosis technology to AUVs. In this study, a system that can monitor the health of in situ AUV thrusters using a convolutional neural network (CNN) was developed. As input data, an acoustic signal that comprehensively contains the mechanical and hydrodynamic information of the AUV thruster was adopted. The acoustic signal was pre-processed into two-dimensional data through continuous wavelet transform. The neural network was trained with three different pre-processing methods and the accuracy was compared. The decibel scale was more effective than the linear scale, and the normalized decibel scale was more effective than the decibel scale. Through tests on off-training conditions that deviate from the neural network learning condition, the developed system properly recognized the distribution characteristics of noise sources even when the operating speed and the thruster rotation speed changed, and correctly diagnosed the state of the thruster. These results showed that the acoustic signal-based CNN can be effectively used for monitoring the health of the AUV’s thrusters.

## 1. Introduction

Needs for ocean exploration and investigations are ever increasing since realizing that the ocean accounts for economic trade carbon storage and oxygen production [1,2]. Furthermore, one-third of the world’s population lives within 100 km of coastlines, where dependence on health and sustainability of the ocean is high [3]. Therefore, the values of oceanographic data in a vast ocean in different time and space zones have become high. Yet, only 5% of the world’s ocean is investigated in some detail [4].

The advent of underwater vehicles that can operate autonomously, autonomous underwater vehicles (AUV), has enabled significant improvements in ocean explorations [5]. As AUVs can operate with minimal human resources and surface vessel supports, they can operate longer in larger regions compared to other underwater platforms, which makes them promising technologies. The AUV platforms allow for low-cost and efficient oceanographic data collection.

However, in such long-duration, large-area missions, the complexity of underwater missions and the possibility of accidents increases. Many oceanographic processes of interest take place in hazardous locations, i.e., shelf breaks, reef structures, etc. [6]. In such conditions, the ability of in situ recognition and evaluation for the failure of hardware components to decide the continuation of the mission becomes essential to recollect the hardware and data recordings. Among the various components, the thruster is one of the most important hardware that could degrade the functionality of the AUV significantly. Recently, research on how to use small micro-robots in the marine environment is being actively studied [7,8].

Traditionally, the fault diagnosis technologies are approached by fusing the on-board data such as yaw angle, surge speed, thruster control voltage, and other conventional sensor readings with two main stages: the feature extraction and the fault identification [9,10]. The data-driven-based approach to extracting fault feature from dynamic signals in time and frequency domains were performed with various feature extraction methods [11,12,13]. For fault identification, model-based classification techniques are commonly used: gray relational analysis (GRA), hidden semi-Markov model, weighted K-nearest neighbor classification, etc. [10,14,15,16]. Moreover, after the advent of machine learning, several methods have been proposed for fault diagnosis: support vector machines [17,18,19,20], artificial neural networks (ANNs) [21], and deep learning [22].

Most of the previous studies were constructed from the data-fusion of dynamic signals available from conventional sensor readings although the methods to extract features and identify the faults are shifted to adopt machine learning [23]. Some researchers have newly adopted vibration signals to detect motor and gearbox faults to account for thruster failures [24,25] since the vibration signals would better represent such defects.

In this paper, an enhanced convolutional neural network (CNN) method for in situ fault classification health monitoring systems for the thruster of the AUV is proposed using the acoustic signals in an embedded system. The acoustic signal is adopted since the underwater radiated noise of the thruster can represent its geometric and hydrodynamic characteristics in operation. The enhanced CNN incorporated acoustic signal manipulation considering the characteristics of the thruster noise signals.

## 2. Materials and Methods

### 2.1. Fault-Type Classification

Figure 1 shows the target model for the development of the AUV health monitoring system in this study. The AUV was modeled concerning the shape disclosed by Dynautics and has the characteristics of a general man-portable AUV. There are four fixed fins on the bow and four controllable fins on the stern, and the thruster is surrounded by a guard structure. The thruster was used by scaling the E779A shape developed by INSEAN. The detailed specifications of the AUV and the thruster are shown in Table 1.

The classification of thruster fault types to be recognized by the AUV health monitoring system developed in this study is shown in Figure 2. In the case of a general underwater thruster, the closer to the tip, the faster the blade rotation speed and the higher the load. In particular, since the thickness of the blade also becomes thinner as it goes from the root to the tip, there is a higher probability of local damage at the tip than at the root. To reflect these practical factors, the damaged thruster geometries were manufactured, while the thruster breakage was limited near the tip. In the case of single-blade damage, it was assumed that the tip part of one blade was slightly broken. Tip breakage was implemented in a direction perpendicular to the blade reference line at a distance of $0.0125\;D_{p}$ from the tip of the blade. In the case of double-blade damage, it was assumed that an additional, more serious failure occurred due to continued operation after single-blade damage. The second tip breakage was set to form 30 degrees with the blade reference line at a distance of $0.025\;D_{p}$ from the tip.

### 2.2. Flow Noise Simulation

Although current signals or vibration signals measured through an accelerometer are often used as inputs of the health monitoring system, this study intends to recognize thruster failures based on the flow noise signals generated by AUV itself. In the case of vibration signals, an accelerometer that can be installed at an appropriate location inside the AUV is additionally required, and a corresponding additional power supply and processing device are also required. On the other hand, in the case of the self-noise signal, the pressure sensor mounted on the outside of the AUV hull for external communication can be used for noise measurement as is.

Blade passing frequency (BPF) noise corresponds to tonal noise with a distinct frequency component and represents the highest noise level among the flow noise sources generated by the thrusters. Furthermore, if the shape of several blades constituting the thruster does not completely match, the noise level increases rapidly and the characteristics of the BPF noise change. In this study, paying attention to this phenomenon, a system that can monitor the health of the thruster through flow noise was proposed. As the demand for missions and coverage for AUVs has increased in recent years, the depth of operation of AUVs has steadily increased as well. Since the cavitation phenomenon is suppressed as the operation depth increases, a non-cavitation condition is assumed in this study.

To secure the self-noise data generated by the AUV thruster as an analytical method, computational fluid dynamics (CFD) and acoustic analogy were used in this study. Figure 3 shows the configuration of the analysis domain for CFD simulation. The overall size of the analysis domain was set to 8 m × 4 m × 4 m in length, height, and width. The left and right boundaries of the domain were set with velocity inlet and pressure outlet conditions, respectively, and symmetry conditions were applied to the remaining boundaries. The configuration of the grid for CFD simulation generated in these domains is shown in Figure 4. For the entire domain, a trimmed mesh corresponding to a three-dimensional unstructured type was applied. As shown in Figure 5, a polyhedral mesh is applied to the area around the thruster. To simulate the rotation of the thruster, the region around the thruster was locally divided and a moving mesh technique was applied.

Table 2 shows the numerical settings applied to the CFD analysis. The target operating speed of the AUV was set to 1 m/s, and the rotational speed that satisfies the self-propelled condition in which the model resistance and the thrust match at that speed were derived (Ω = 27 rps). To realize the unsteady characteristics of the flow field due to the rotation of the thruster, the time step was set small enough so that the rotation angle of the time interval did not exceed 5 degrees. As the turbulence model, RANS k-ω, sst was applied. To derive the flow noise generated from UUV, an acoustic analogy was applied. The acoustic analogy is the most common and widely used theory to calculate flow noise in aeroacoustics and is actively applied in the field of hydroacoustics. The acoustic analogy is a concept that extracts the corresponding flow noise source based on the hydrodynamic properties derived through CFD analysis. The Ffowcs Williams and Hawkings (FW-H) equation analogizes the noise generated when an arbitrary-shaped solid moves in a flow field as the source term of the wave equation. In general, a hybrid method that extracts flow field information through incompressible CFD analysis and calculates a noise signal as a flow noise source derived by applying acoustic analogy is widely used. In order to calculate the noise measured by a specific receiver, Green’s function must be applied to the FW-H equation, and the noise source must be calculated through surface integral and volume integral. For this calculation, numerically advantageous formulas have been developed through various modifications of the FW-H equation, and a representative is Formulation 1A [26]. In this study, the flow noise generated from AUV was calculated by applying Formulation 1A.

### 2.3. Data Acquisition

To develop a health monitoring system, it is essential to secure a lot of data for each type of fault classified in Figure 2. In the data-driven method, as a large amount of fault data are used, the range that the developed monitoring system can cover expanded, and the accuracy of fault identification increases.

The procedure for acquiring failure data applied in this study is shown in Figure 6. First, using flow noise simulation, data on flow noise generated in AUV in normal and faulty conditions are derived. As mentioned above, since the data-driven method has the characteristic of increasing accuracy according to the absolute amount of failure data, artificially augmenting the given flow noise data is advantageous for system development. For one-dimensional signals like flow noise data, data can be augmented by adding white Gaussian noise to the original signal. To add noise, the signal to noise ratio (SNR)—which determines the ratio of the original signal to the noise, as expressed in (Equation 1)—is introduced. To multiply the data, while preserving the acoustic information included in the original signal, the SNR was set to a random number between 68 and 73 dB, and data corresponding to 10 times the original signal was additionally secured.
(1)$$SNR_{dB}=10log_{10}\left(\frac{P_{signal}}{P_{noise}}\right)$$

The derived data are expressed in the form of a one-dimensional sequential signal. These can be directly applied to the development of health monitoring systems, but in order to increase the efficiency of the system, it is necessary to pre-process the data in a form that can express the physical characteristics included in the input signal. In this study, the one-dimensional noise signal is converted into two-dimensional image data by using wavelet transform. In the case of the wavelet transform, one-dimensional data are expanded into a two-dimensional image, that has the advantage of being able to contain transient information about time while transforming it into a frequency domain. Therefore, when the wavelet transform is applied to the inputted flow noise signal, two-dimensional image data containing the change in the frequency characteristic of the AUV noise with time are finally derived.

### 2.4. Continuous Wavelet Transform

The continuous wavelet transform is similar to the Fourier transform, but since it can additionally secure the time-frequency resolution, it is widely used to investigate the time-varying frequency spectrum.
(2)$$F\left(a,b;f\left(t\right),\psi \left(t\right)\right)=\frac{1}{\sqrt{a}}\int_{-\infty }^{\infty }f\left(t\right)\psi^{*}\left(\frac{t-b}{a}\right)dt,a>b\in R$$ Equation (Equation 2) represents the continuous wavelet transform for the input signal f(t). Here, *a* and *b* mean the scale and translation parameters of the wavelet transform, respectively. ψ(t) stands for a mother wavelet function that is continuous in both the time and frequency domains. A daughter wavelet is generated as a result of the mother wavelet being adjusted by the appropriate scale and translation parameters. * denotes the complex conjugate. The mother wavelet used in this study is a generalized-Morse wavelet, and is expressed as (Equation 3) by taking a Fourier transform. Here, U(w) represents the Heaviside step function, and $a_{\beta ,\gamma }$ represents the normalizing constant expressed as in (Equation 4) [27].
(3)$$\Psi_{\beta ,\gamma }\left(\omega \right)=U\left(\omega \right)a_{\beta ,\gamma }\omega^{\beta }e^{-\omega^{\gamma }}$$
(4)$$a_{\beta ,\gamma }\equiv 2{\left(e\gamma /\beta \right)}^{\beta /\gamma }$$

### 2.5. Deep Convolutional Neural Network for AUV Health Monitoring

A typical deep neural network (DNN) consists of only fully connected layers (FCLs), and targets one-dimensional array data. In order to process image data using DNN, image data composed of three dimensions must be flattened into one dimension, and in this process, spatial information contained in the image is lost. A model to compensate for these problems is the CNN.

Since CNN receives an image as input and applies convolution to build up a layer of features, spatial information included in the input data is not lost, and information about the association between adjacent pixels can be effectively recognized. Figure 7 shows the general architecture of CNN. It can be divided into a section for extracting image features and a section for classifying the derived features. The feature extraction section is composed of a pair of convolution layers and pooling layers stacked in multiple layers. In the convolution layer, an activation function is applied after convolution of a given filter to the input image. The pooling layer maximizes the features included in the data by selectively maintaining or discarding the derived features. In the classification section, the data derived through feature extraction is flattened and converted into a one-dimensional array, and then consists of a number of FCLs similar to the general DNN configuration for image classification.

#### 2.5.1. Convolution Layer

The convolutional layer corresponds to the main part constituting the feature extraction stage of CNN. Convolution simply refers to the operation of applying a filter to input data to obtain an activated result. By iteratively performing convolution, a distribution of activated features can be obtained, which is called a feature map. Convolution is a linear operation of the dot product of input data and a package of weights called filters or kernels. The filter is smaller in size than the input, and is superimposed on the image and moves in the vertical and horizontal directions to derive one scalar value through a dot product. As the image iteratively passes through the convolutional layer, the resulting feature map shrinks in size and information about the pixels located at the edges is lost. To prevent data loss, zero-valued pixels are added to the edges of the image, called zero-padding.

#### 2.5.2. Activation Function

In feature extraction of CNN, the signal derived after performing convolution is transmitted to the next layer through the activation function. A neural network with multiple hidden layers with no activation function applied is equivalent to a simple linear classifier. When a nonlinear activation function is applied to a neural network, the complexity is increased, and the nonlinear problem-solving ability of the neural network can be obtained. In this study, the rectified linear unit (ReLU) function expressed as (Equation 5) was applied. In the case of sigmoid or hyperbolic tangent activation functions, a gradient vanishing problem occurs in the backpropagation process for weight evaluation. However, the ReLU function is suitable as an activation function for deep neural networks because gradient vanishing does not occur [28].
(5)$$h\left(x\right)=x^{+}=max\left(0,x\right)$$

#### 2.5.3. Dropout Layer

If multiple filters are applied and deep layers are stacked to improve the performance of the neural network, the number of hyperparameters that determine the characteristics of the neural network becomes very large. When the number of parameters of the neural network is too large, a problem of overfitting input data occurs during neural network training. To solve this problem, Hinton et al. [29] proposed a dropout layer. Dropout is a concept that randomly turns off the nodes constituting the FCL with a set probability between 0 and 1. When dropout is applied, it is possible to prevent the problem of over-concentrating only the features that occur in a specific node and interpreting it and to derive an unbiased output value. This concept is applied to AlexNet and it has been proven that the performance of neural networks is improved [30].

#### 2.5.4. Maxpooling

Pooling is a process of sub-sampling for feature maps derived from convolutional layers. The size of the data can be reduced through pooling, and is effective in solving the overfitting problem by reducing the number of parameters. In this study, max pooling, which samples the maximum value at a given size, was applied.

#### 2.5.5. Global Average Pooling

Global average pooling (GAP) is a type of pooling, and is a representative method to prevent overfitting. When the number of filters used in CNN is large, there are also many feature maps in the output, resulting in a problem in which the dimensionality of the neural network increases and the number of parameters rapidly increases. To solve this problem, Lin et al. [31] derives a one-dimensional vector corresponding to the number of channels by substituting the total average value of features derived from one channel. This method can be utilized by replacing the FCL in the final output, and in some cases, the performance of the neural network can be improved through the combination of GAP and FCL.

## 3. Results

### 3.1. Optimizing the Deep Convolutional Neural Network

In this study, a monitoring system that can evaluate thruster health based on the flow noise signal generated by AUV is constructed using CNN. For this, an appropriate CNN architecture setting and optimization process are required. In this study, the final CNN architecture is determined by composing two types of artificial neural networks and comparing their accuracy.

#### 3.1.1. Architectural Optimization of Convolutional Neural Networks

In general, it is known that the depth and performance of artificial neural networks have a positive correlation. However, when an excessively deep neural network is constructed, the number of hyperparameters that determine the characteristics of the neural network increases, and a problem of overfitting may occur. Therefore, it is essential to find the optimal layer through various attempts to construct an appropriate neural network.

In this study, as shown in Figure 8, two types of neural networks were created and the accuracy was compared. In the case of Type-A, filters having a size of 10 × 10 were used, and each of the four layers consisted of a convolution layer and a ReLU function. The number of filters used for each layer is sequentially 8, 8, 16, and 32. The derived feature map is converted into a one-dimensional vector through flattening. After that, the final output was derived through the FCL and the softmax layer. In the case of Type-B, filters with a size of 5 × 5 that are smaller than those of Type-A were applied, and each of the six layers was composed of a combination of a convolution layer and a ReLU function. The number of filters used for each layer is sequentially 8, 16, 16, 32, 64, and 64. After that, by applying GAP, the feature map is converted into a one-dimensional vector having a length of 64 and passed through a layer composed of a combination of an FCL and a dropout layer. In the subsequent step, the final output was derived after going through the softmax layer in the same way as in type-A. The size of the input image used in this study is 400 × 400 × 3. For the training set, 70% of the data were randomly selected from the entire data set and the remaining 30% of the data were assigned to the test set.

Unlike Type-A, Type-B minimizes the size of the filter to reduce memory, but increases the number of filters used and extends the depth of the neural network. In this case, the problem of overfitting may occur as the neural network deepens. To prevent this, in Type-B, a combination of GAP, FCL, and Dropout was applied instead of composing the classification layer only with FCL. In particular, GAP is known to be effective in processing high-dimensional feature maps derived from a large number of filters, so it is appropriate to apply to Type-B.

#### 3.1.2. Comparison of Trained Neural Networks of Two Architecture Types

By training two neural networks designed with different concepts, the accuracy and overfitting were compared and analyzed from the viewpoint of constructing a health monitoring system. For both Type-A and Type-B, the training settings were the same as in Table 3. Figure 9 shows the accuracy and loss functions derived from training. In the case of Type-A, the accuracy and loss values fluctuate heavily in the section where the iteration is less than 20, but it is relatively stabilized when the iteration exceeds 20. However, between 20 and 100 iterations, the accuracy partially decreases and the loss value partially increases. A characteristic of well-structured neural network training is that the accuracy continuously increases with iteration and the loss continuously decreases. During the training of the neural network, if the loss function continues to decrease and then bumps appear as the trend changes in a specific section, it can be estimated that overfitting has occurred. In the case of Type-A, it can be seen that some overfitting occurred in the middle. Conversely, in the case of Type-B, it can be seen that both accuracy and loss converge well without any fluctuations in the entire iteration section. In particular, even in the case of loss, it shows a tendency to continuously decrease without a bump section in the middle.

Table 4 compares the final accuracy and loss after training for two types of neural networks. As can be expected from the trend shown in Figure 9 above, it can be seen that Type-B exhibits better accuracy and lower loss than Type-A. By comparing these results, it can be concluded that the neural network architecture of Type-B is more suitable for the health monitoring system than Type-A. All subsequent neural networks were trained using the same architecture as Type-B.

### 3.2. Accuracy Analysis according to Pre-Processing Method

In this section, the effect of data pre-processing on the development of the AUV health monitoring system was analyzed. In the case of flow noise generated in AUV, the level of BPF noise from the thruster is very dominant compared to other noise sources. Furthermore, damaged thrusters generally generate higher levels of BPF noise than thrusters under normal conditions. Considering these acoustic characteristics, it can be expected that the accuracy and utility of the derived health monitoring system may vary depending on how the collected flow noise data is pre-processed.

Therefore, in this section, various modifications of the pre-processing technique applied to the flow noise data were analyzed, and the change of the monitoring system was analyzed accordingly. Basically, the data processing technique is common to wavelet transform, but various modifications can be applied in terms of acoustic scale. In the acoustic scale, there is a choice of whether to use the flow noise data as a linear pressure scale or a dB scale, sound pressure level (SPL). In addition, there is a way to normalize the noise data to the maximum value. In particular, as mentioned above, the failure is likely to be identified simply by an increase in the BPF noise level, since the damaged thruster generates a higher-than-normal flow noise. When the neural network is trained with data normalized to the maximum value of noise, it can be expected that the neural network will diagnose the failure by recognizing the noise distribution characteristics rather than information about the noise level.

Therefore, in this study, three different data pre-processing methods were selected as follows:Linear Pressure;Sound Pressure Level based on Decibel Scale;Sound Pressure Level Normalized by Maximum Noise.

#### 3.2.1. Linear Scale

First, Figure 10 shows the comparison of the scalograms that appear in the normal and damaged conditions of the thruster. In all three cases, a high noise level occurs at the first BPF (4 × 27 = 108 Hz), which corresponds to the product of the number of blades (B=4) and rotational speed (Ω = 27 rps). Since the noise level is expressed on a linear scale, there is a disadvantage that only the BPF noise is emphasized in the distribution characteristic of the flow noise. In particular, in the case of a thruster propulsion model such as the target AUV, since BPF noise is dominant in the generated flow noise, the distribution characteristics of other noise sources are masked by the BPF noise. Unlike the decibel scale, in the linear scale, other noise sources except for the BPF noise are expressed at an excessively low level. Therefore, in the image data pre-processed on the linear pressure scale, the change in the BPF noise level due to thruster damage is most emphasized, and the change in the distribution characteristics of other noise sources is not prominent.

#### 3.2.2. Decibel Scale

Figure 11 shows the scalogram of flow noise expressed in decibel scale by comparing the thruster state. Unlike the previous case, since decibel based on log scale was applied, it can be confirmed that noise sources lower than BPF noise are properly decomposed. Again, it can be seen that a high noise level appears in the first BPF noise. However, unlike the previous linear scale, the decibel scale implements the distribution of flow noise with log-scale, revealing the distribution characteristics of other noise sources in addition to the BPF noise. When looking at each noise source component, since BPF noise has a high level, the distribution characteristic appears mainly in the red component, and other noise sources are distributed in the green and blue components. However, the change in the level of BPF noise due to thruster damage is still the most prominent throughout the image.

#### 3.2.3. Normalized Decibel Scale

Figure 12 shows the result of normalizing the distribution of flow noise to the maximum noise value. The noise distribution data were normalized by dividing the noise level across the scalogram by the maximum value of noise derived from each scalogram, and then unifying the maximum value to 86 dB for all data. Through this process, the significant change in BPF noise level due to thruster damage, which appeared in the previous technique, was compensated for. As can be seen from the comparison of the noise distribution results, since the maximum values of all scalograms are unified, there is no change in the level of BPF noise depending on the damage condition of the thruster. Instead, it can be seen that the distribution characteristics of noise sources other than BPF noise are actively changing depending on the damage state of the thruster. This feature is expected to help the neural network not only focus on the BPF noise source in the learning process, but comprehensively recognize the distribution characteristics of other noise sources.

#### 3.2.4. Comparative Analysis of Neural Networks According to Pre-Processing Method

Table 5 shows the configuration of the data set used to train the neural network. The training set and validation set were divided in a ratio of 7 to 3, and training was performed using 704, 640, and 560 data for the normal condition, single-blade damage, and double-blade damage conditions, respectively. Table 6 compares the accuracy and loss derived when validating three neural networks trained using data from different pre-processing methods. First, in that all three neural networks showing a high accuracy of over 99%. It can be said that the effectiveness of the health monitoring system of the AUV thruster developed in this study has been secured. Furthermore, there was a slight difference in accuracy depending on the pre-processing technique. The two neural networks using the decibel scale showed 100% accuracy, whereas the neural network applying the linear scale showed a slightly lower accuracy of 99.90%. As shown in the comparative analysis of the pre-processed images, in the case of the linear scale, it is difficult to examine the distribution characteristics of other noise sources except for the BPF noise. As a result, it is expected that the decrease in accuracy is caused by the tendency of the neural network to over-recognize the level of BPF noise during the learning process. Even in the normal condition of the thruster, flow noise similar to that of a single-blade damage condition rarely occurs, so recognizing a failure based on the level of the BPF noise alone may act as a factor to lower the accuracy.

### 3.3. Performance of AUV Health Monitoring System for Off-Training Conditions

As mentioned above, the developed neural networks exhibit high identification accuracy close to 100% under the learned conditions. In this section, the classification accuracy was analyzed for the derived neural networks under different conditions from the data used for learning. The off-training conditions selected for the test are as follows.

Normal thruster, Moderate operation speed ($U_{moderate}$);Normal thruster, Rapid operation speed ($U_{rapid}$);Single-blade damaged thruster, Silent operation speed ($U_{silent}$).

The purpose of performing tests on off-training conditions is to analyze the potential performance and scalability of the trained neural network. In particular, it was analyzed whether the developed system could capture and accurately classify the characteristics of the potentially implied noise distribution based on the state of the thruster, even for data not used for training. As mentioned earlier, damaged thrusters typically produce higher noise levels than normal thrusters. In addition, the faster the thruster rotates, the higher the noise level it generates. Therefore, for the off-training test, a condition that can be as confusing as possible for the neural network to classify noise data was derived. As a result, two conditions were selected that generate high noise similar to the damaged thruster due to the thruster being in a normal state but operating at a high speed. In addition, although the thruster was damaged, a single condition—in which low noise similar to that of a thruster in a normal state was generated due to a low operating speed—was selected. Table 7 provides information on thruster conditions and operating speeds for the selected off-training test conditions. For the first two conditions of moderate and rapid operation, the thruster was in normal condition and the operating speed was increased from 1.0 m/s to 1.2 m/s and 1.5 m/s, respectively. As the operation speed of the AUV increased, the self-navigation point also changed accordingly, and the thruster rotation speed increased from 27 rps to 32 rps and 37 rps in moderate and rapid operation, respectively. In the damaged thruster condition, silent operation was applied to reduce the operating speed to 0.7 m/s, and accordingly, the self-navigation point also changed, reducing the thruster rotational speed from 27 rps to 23 rps.

Figure 13 shows scalograms of flow noise expressed in linear scale generated under off-training conditions. Figure 14 and Figure 15 show scalograms of the same flow noise expressed in decibel scale and normalized decibel scale, respectively.

The normal-moderate conditions ($U_{moderate}$ = 1.2 m/s) are faster than the cruising speed ($U_{cruising}$ = 1.0 m/s) and the thruster rotation speed increases, so it can be seen that a higher level of BPF noise is generated at a BPF (4×32 = 128 Hz) higher than the BPF (4×27 = 108 Hz) at the cruising speed. However, since the operating speed is not significantly different, the noise distribution result is not significantly different from the normal condition at the cruising speed. In normal-rapid conditions, since the rapid operating speed ($U_{rapid}$ = 1.5 m/s) is faster than moderate ($U_{moderate}$ = 1.2 m/s), it can be seen that the first BPF (4×37 = 148 Hz) is further increased and the noise level is also higher. In the case of single-blade damage-silent conditions, it can be confirmed that the speed ($U_{silent}$ = 0.7 m/s) is slower than the cruising speed ($U_{cruising}=$ 1.0 m/s) and the rotation speed of the thruster is reduced, so that the BPF noise is generated at a lower frequency (4×23 = 92 Hz) than the cruising condition.

In the case of the results expressed in linear scale in Figure 13, as mentioned above, the BPF noise is overly emphasized, and the distribution of other noise sources is not revealed. Furthermore, in the case of the normal–rapid condition, it will be difficult for the neural network to properly classify it as a normal condition because the level of BPF noise appears similar to that of the single-blade damaged condition in Figure 10 even though the thruster is in a normal condition. In the case of the results expressed in the decibel scale in Figure 14, the distribution of noise sources other than BPF noise is also revealed. However, in the case of the normal–rapid condition, it can still be confirmed that it appears similar to the noise distribution characteristic shown in the single-blade damaged condition of Figure 11.

On the other hand, in the case of the result expressed in the RGB-normalized dB scale in Figure 15, the maximum value is normalized and the distribution characteristics of the entire noise sources are emphasized. Accordingly, it can be seen that the result of the normal–rapid condition shows a distribution characteristic similar to that of the normal condition in Figure 12. Therefore, it seems that the neural network can properly classify normal–rapid results into normal conditions.

Table 8 shows the accuracy derived by performing the test in the off-training condition of the neural network. Three neural networks using different pre-processing methods were used. As mentioned above, in the case of normal–moderate conditions, the moderate speed did not differ significantly from the cruising speed, so the distribution or magnitude of the BPF noise did not change much. Accordingly, 87.5% accuracy was obtained in all three neural networks. On the other hand, in the case of the normal–rapid condition, it is very difficult to properly classify the results because they are very similar to the results under the single-blade damage condition at cruising speed. Therefore, neural networks applied with linear scale and decibel scale show a low accuracy of 0%. On the other hand, the normalized dB scale showed better results than other neural networks with 50% accuracy as it focused on recognizing the distribution of noise sources by applying normalization. In the case of the single silent operation, the neural network to which the linear scale was applied showed an accuracy of 66.7%, but both neural networks to which the decibel scale was applied showed an accuracy of 100%. This seems to indicate that the level of BPF noise was overemphasized in the learning of the neural network to which the linear scale was applied, and an error of mistaking the level of BPF noise generated in single-silent operation as a normal state appeared.

## 4. Conclusions

In this study, a health monitoring system was developed for AUV thrusters using artificial neural networks. As input data for health monitoring, self-flow noise measured in AUV was adopted. By converting the flow noise, which is a one-dimensional array, into two-dimensional data through continuous wavelet transformation, an image that can analyze not only frequency characteristics but also transient components was derived. Although there are various types of neural networks, CNNs, which are known to be suitable for analyzing spatial correlation information contained in 2D image data, have been used.

First, to optimize the CNN architecture, two neural networks with different filter sizes and different layer depths were designed, while maintaining the number of hyperparameters. As the neural network deepens, overfitting may occur due to an increase in the number of hyperparameters. To solve this problem, a combination of GAP and Dropout layers was used. Constructing a deeper layer with a relatively small filter size showed excellent results in terms of accuracy and loss.

Afterward, the performance of neural networks was compared by different pre-processing techniques. Three pre-processing techniques were selected: a method of expressing linear pressure, a method of expressing decibel-based SPL, and a method of expressing normalized SPL. When linear pressure was applied, the level of the BPF noise source was too high, and all other noise sources were masked. Thus, the decibel scale was superior in terms of the resolution of the noise source distribution. When normalization is applied, the distribution of noise sources is emphasized rather than focused only on changes in the level of BPF noise. As a result of training, all three techniques showed high accuracy, but the decibel scale showed superior performance as a pre-processing technique than the linear pressure scale.

Off-training conditions were also tested on neural networks. The ’normal–moderate’ and ’normal–rapid’ conditions in which the operating speed is increased under the normal thruster condition and the ’single–silent’ condition in which the operating speed is decreased under the single-blade breakage condition were set. As a result of testing the neural networks derived for three off-training conditions, all three neural networks generally achieved high accuracy in the ’normal–moderate’ and ’single–silent’ conditions. However, in the case of the ’normal–rapid’ condition, the BPF noise level was very similar to the level shown in the ’single-cruising’ condition, so very low accuracy was derived from the neural networks of the linear scale and decibel scale. On the other hand, in the case of the normalized dB scale, the best accuracy was derived because it is based on the distribution characteristics of noise sources other than BPF noise. Through this, it was confirmed that the performance and application range of the neural network for health monitoring can be expanded depending on the data pre-processing technique.

In the future, by expanding this study, rather than based on single input data, a health monitoring system, including not only the thruster but also the hull and appendages of the AUV, is planned to be developed through the correlation of data from multiple sensors.

## Figures and Tables

**Figure 1 sensors-22-07073-f001:**
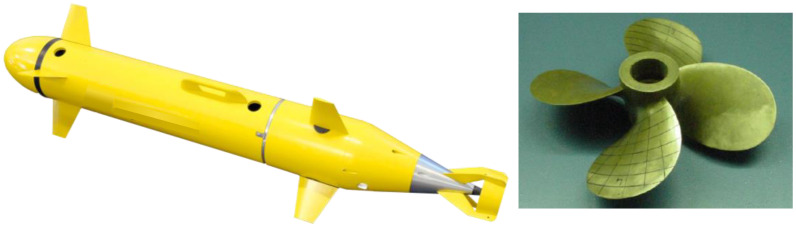
Geometries of AUV and thruster. (**left**) AUV; (**right**) E779A Propeller.

**Figure 2 sensors-22-07073-f002:**
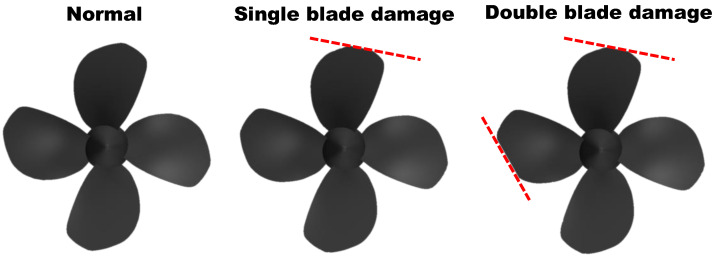
Configuration of thruster fault types. (**left**) Normal; (**middle**) single-blade damage; (**right**) double-blade damage.

**Figure 3 sensors-22-07073-f003:**
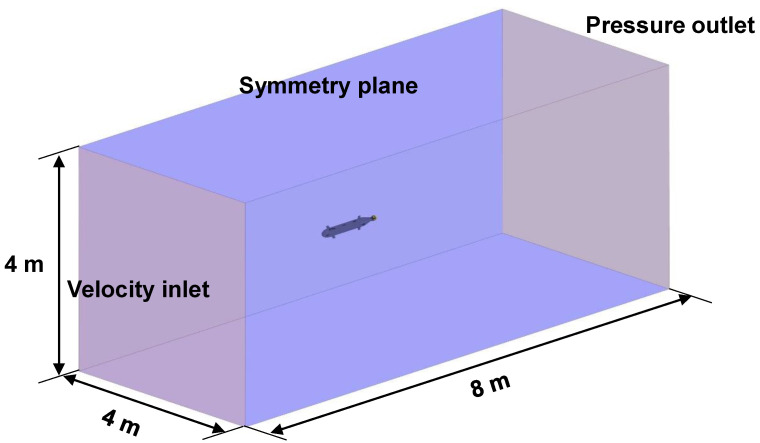
Specification of simulation domain and boundary condition setting.

**Figure 4 sensors-22-07073-f004:**
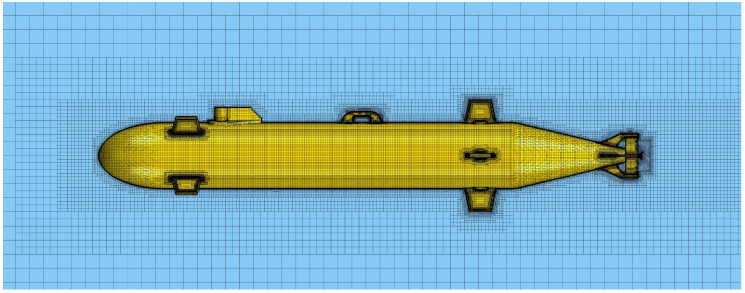
Mesh configuration of AUV.

**Figure 5 sensors-22-07073-f005:**
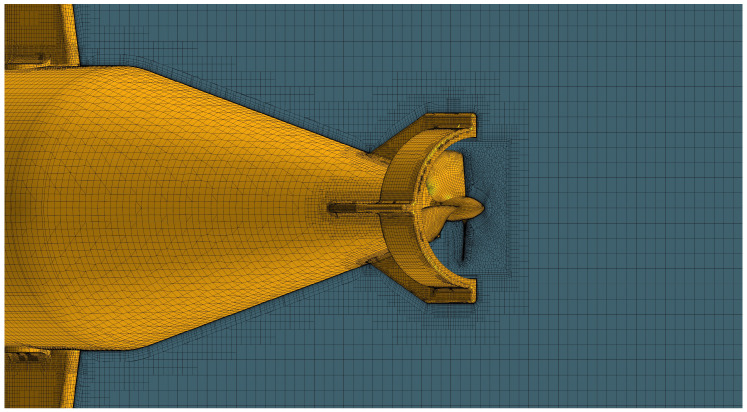
Mesh configuration around the thruster.

**Figure 6 sensors-22-07073-f006:**
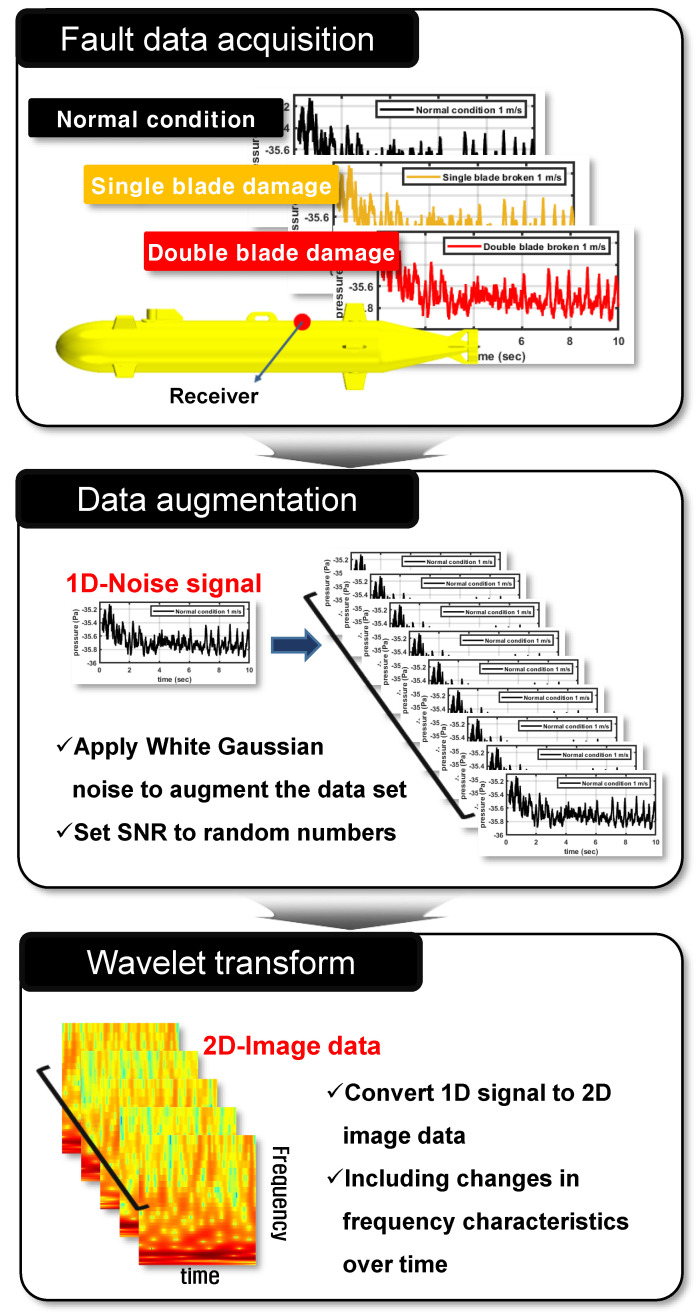
Procedure of the data acquisition process.

**Figure 7 sensors-22-07073-f007:**
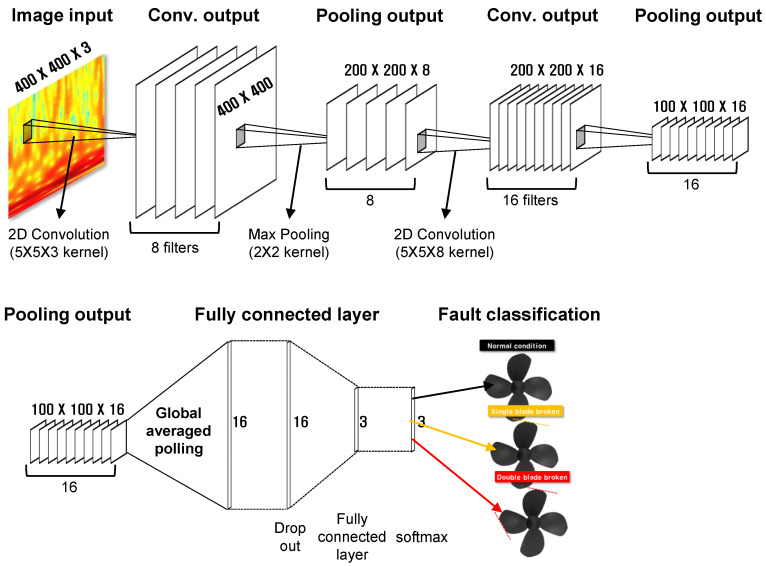
CNN architecture for AUV health monitoring. (**top**) Feature extraction; (**bottom**) Classification.

**Figure 8 sensors-22-07073-f008:**
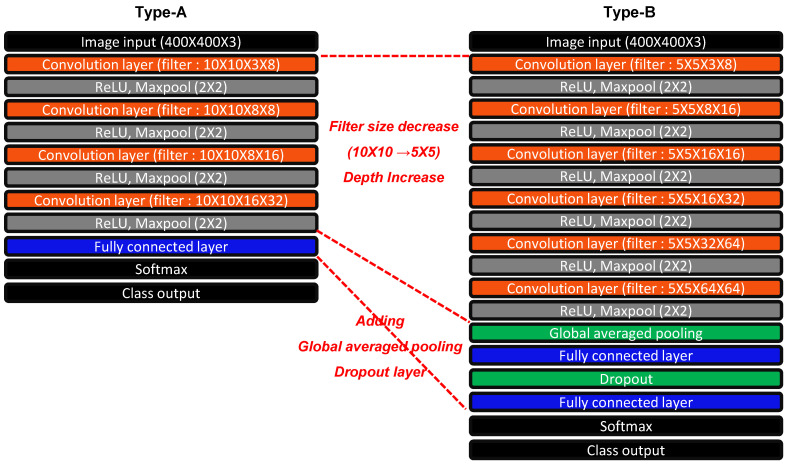
Comparison of two types of CNN architecture configurations.

**Figure 9 sensors-22-07073-f009:**
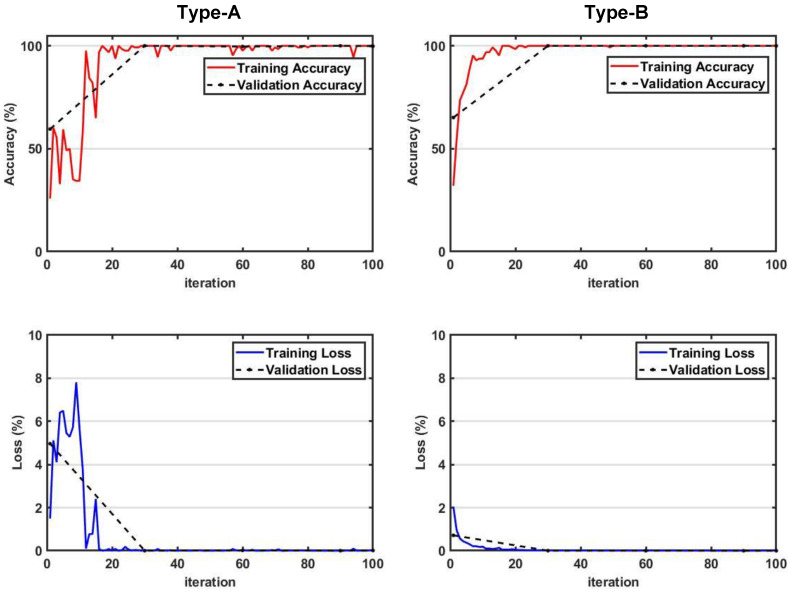
Comparison of accuracy and loss function for the training of two types of CNNs. (**top** line) Accuracy; (**bottom** line) Loss.

**Figure 10 sensors-22-07073-f010:**
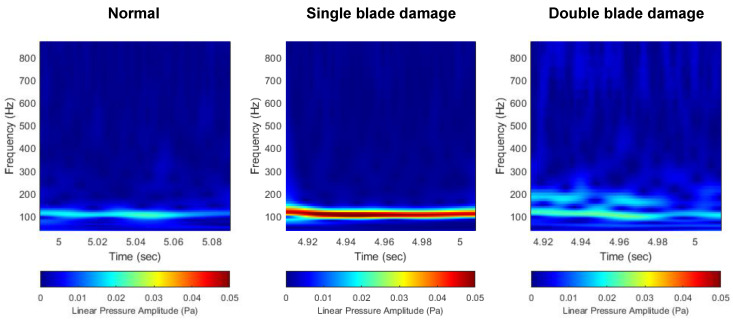
Comparison of flow noise scalogram with linear scale. (**left**) Normal, (**middle**) single-blade damage, (**right**) double-blade damage.

**Figure 11 sensors-22-07073-f011:**
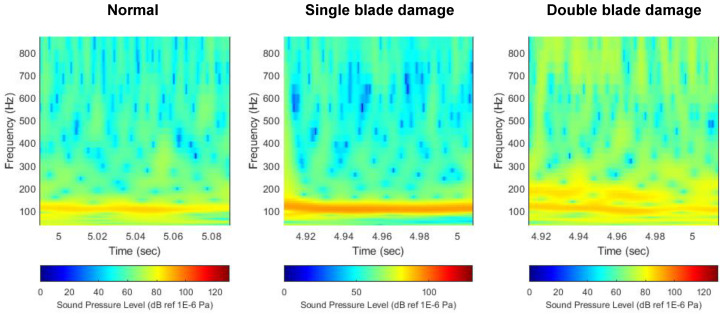
Comparison of flow noise scalogram with decibel scale. (**left**) Normal, (**middle**) single-blade damage, (**right**) double-blade damage.

**Figure 12 sensors-22-07073-f012:**
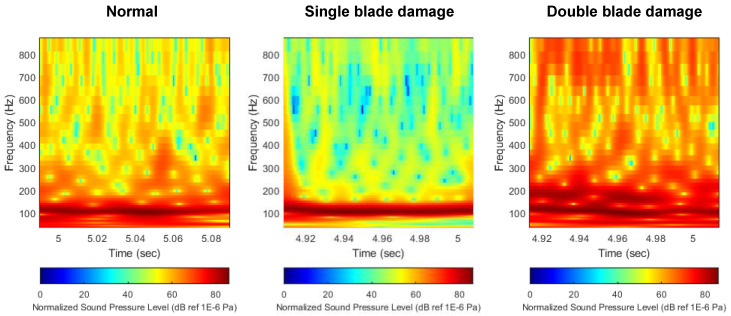
Comparison of flow noise scalogram with normalized decibel scale. (**left**) Normal, (**middle**) single-blade damage, (**right**) double-blade damage.

**Figure 13 sensors-22-07073-f013:**
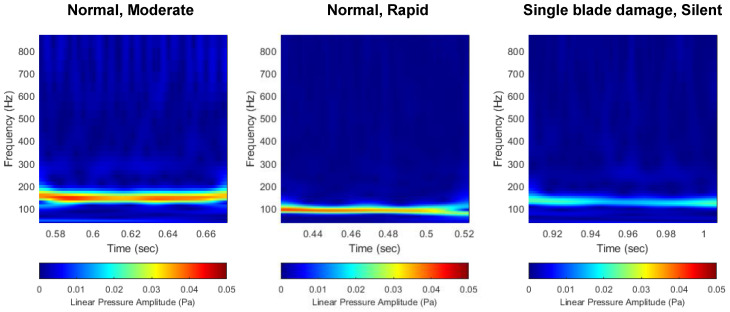
Flow noise scalogram for off-training condition with linear scale. (**left**) Normal–moderate; (**middle**) normal–rapid; (**right**) single-blade damage-silent.

**Figure 14 sensors-22-07073-f014:**
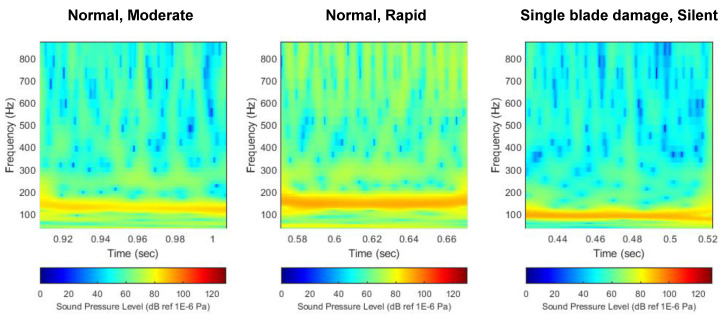
Flow noise scalogram for off-training condition with decibel scale. (**left**) Normal–moderate; (**middle**) normal–rapid; (**right**) single-blade damage-silent.

**Figure 15 sensors-22-07073-f015:**
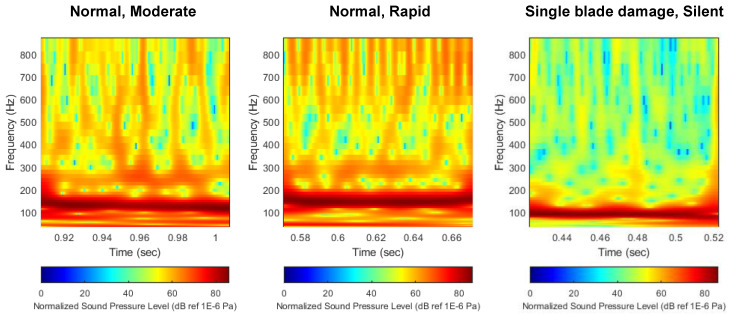
Flow noise scalogram for off-training condition with normalized decibel scale. (**left**) Normal–moderate; (**middle**) normal–rapid; (**right**) single-blade damage-silent.

**Table 1 sensors-22-07073-t001:** Specifications of AUV and thruster.

Model	Description	Symbol	Dimension
AUV	Length	*L*	1.3 m
	Maximum hull radius	$R_{max}$	80 mm
	Weight	*W*	12 kg
Thruster	Blade number	*B*	4
	Diameter	$D_{p}$	0.064 m
	Pitch ratio	$P/D_{p}$	1.1

**Table 2 sensors-22-07073-t002:** Numerical settings for flow noise simulation.

Settings	Description
Time step (Δt)	5 × $10^{-4}$ s
Cruising speed ($U_{cruising}$)	1 m/s
Thruster rotation speed (Ω)	27 rps
Turbulence model	RANS k-ω SST
Mesh elements	3D unstructured, Trimmer mesh
Solver	3D Unsteady, Segregated
Temporal discretization	1st order implicit
Convection	2nd order
Gradient	Hybrid Gauss-LSQ

**Table 3 sensors-22-07073-t003:** Training options for convolutional neural network.

Type	Settings	Description
Type-A, Type-B	Solver	Stochastic Gradient Descent with Momentum (SGDM)
	Momentum	0.9
	Initial Learn Rate	0.01
	Learn Rate Drop Factor	0.1
	Learn Rate Drop Period	10
	L2 Regularization	1 × $10^{-4}$
	Max Epochs	10
	Mini Batch Size	128
	Validation Frequency	50
	initial Learn Rate	0.01

**Table 4 sensors-22-07073-t004:** Training results for two different types of convolutional neural network.

Type	Accuracy	Loss
Type-A	99.65%	0.784%
Type-B	100.00%	0.034%

**Table 5 sensors-22-07073-t005:** Data set composition in training and validation.

Condition	Training Data Set (70%)	Validation Data Set (30%)
Normal	493	211
Single-blade damage	448	192
Double-blade damage	392	168

**Table 6 sensors-22-07073-t006:** Training results for convolutional neural network with different pre-processing methods.

Pre-Processing	Accuracy	Loss
Linear Scale	99.90%	0.102%
Decibel Scale	100.00%	0.034%
Normalized Decibel Scale	100.00%	0.052%

**Table 7 sensors-22-07073-t007:** Off-training test conditions.

Off-Training	Thruster Condition	Operation Speed	Thruster Rotation Speed
Normal–Moderate	Normal	$U_{moderate}$ = 1.2 m/s	32 rps
Normal–Rapid	Normal	$U_{rapid}$ = 1.5 m/s	37 rps
Single–Silent	Single-blade damaged	$U_{silent}$ = 0.7 m/s	23 rps

**Table 8 sensors-22-07073-t008:** Comparing the accuracy of neural networks in off-training tests.

Pre-Processing	Off-Training Condition	Accuracy
Linear Scale	Normal-Moderate operation	7/8 (87.5%)
	Normal-Rapid operation	0/8 (0%)
	Single-Silent operation	12/18 (66.7%)
Decibel Scale	Normal-Moderate operation	7/8 (87.5%)
	Normal-Rapid operation	0/8 (0%)
	Single-Silent operation	18/18 (100%)
Normalized Decibel Scale	Normal-Moderate operation	7/8 (87.5%)
	Normal-Rapid operation	4/8 (50%)
	Single-Silent operation	18/18 (100%)

## Data Availability

Not applicable.

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
