# Peer review of "Enhanced Convolutional Neural Network for In Situ AUV Thruster Health Monitoring Using Acoustic Signals"

_sensors, 2022, doi:10.3390/s22187073_

Round 1

Reviewer 1 Report

Within the scope of the draft manuscript, the authors developed the health monitoring systems for in-situ AUV thruster by adopting a convolutional neural network to acoustic signals. The authors compared the accuracy of neural networks by applying various pre-processing methods to the acoustic signal. The manuscript deals with health monitoring, which is attracting attention as a core technology of underwater drones, and it is interesting in that the system developed using acoustic signals shows high accuracy. However, some issues must be addressed before the article can be regarded as complete:

Q1. To satisfy the reproducibility, it seems that specific figures and conditions for the damage state of the thruster should be additionally presented.

Q2. It is known that results of CFD analysis depends on the size of the domain and the configuration of the grid. Can it be considered that the results derived from the grid used in this study are converged?

Q3. It is mentioned that the acoustic analogy is applied to calculate the flow noise, but the detailed methodology is omitted. It seems that an additional description of the calculation process of flow noise is necessary.

Q4. It is mentioned that the characteristics of the data recognized by the neural network change depending on the pre-processing method, but it seems that additional explanation is needed. If necessary, how about comparing the feature maps extracted from neural networks for each method?

Reviewer 2 Report

The article focuses on applying deep learning techniques for the truster monitoring of an underwater vehicle. Even though the paper is well-written and carefully prepared, it contains severe flaws in the methodology and should not be published. 

Firstly, the research problem is artificial. For many years of my practice, I have never heard about a partially broken propeller. In some cases, the propeller has been lost, but more often, the bearings have been damaged. Consequently, developing a system which can monitor blade damage in real time and is capable of distinguishing between one or two broken blades is pointless.

Secondly, the presented solution can not be implemented under real conditions. The authors performed a very simple CFD simulation to model the propeller's noise. Unfortunately, noise modelling in the underwater environment poses a very demanding task and only a few applications, which can generate reliable results, are on the market. The better option would be to record noise in the laboratory channels - this procedure is often applied to determine ship propeller characteristics. In this attempt, cavitation, which has a substantial impact on generated noise, is taken into consideration. But the experiments are carried out under laboratory conditions. An AUV cannot perform the same measurement during operation because it generates a lot of noise by itself. What is more, sound propagation in the water is associated with the phenomena such as physical and chemical properties of seawater that cause attenuation and refraction, rough and poorly defined surfaces, and ambient noise and reverberation. As a result, the noise measurement and interpretation are very challenging, especially on a moving AUV.

Finally, the authors generated data for a narrow range of movement parameters (one cruising speed, one thruster rotation speed). It seems to be too few to utilise under real conditions. Additionally,  adding the White Noise to the scarce data as an augmentation step is not a good idea. Data augmentation artificially increases the data size; consequently, finding a pattern such as the White Noise does not pose a demanding task for neural networks.

Reviewer 3 Report

1. The manuscript contains spelling errors, such as the misspelling of the coastline in the first paragraph of the introduction. Please confirm the accuracy of the description in the manuscript and take it seriously.

2. Convolutional neural network in section 2.5 of the manuscript can be shortened or even need not be introduced, because deep learning is a common method in the field of fault diagnosis, the basic structure of CNN has been widely known, and no one will pay attention to this part.

3. Please indicate the specific size of the data set, the data processing method and how the training set and test set are divided in Section 3.1

4. In Section 3.2, the acoustic scale should be the main factor affecting the data quality, not whether the data is RGB format. However, the abstract and the conclusion focus on the picture style. To reach this conclusion, a total of six groups of control experiments with two picture styles and three acoustic scales should be carried out instead of the three groups in the manuscript.

5. According to the statement in Section 3.3, different working conditions of AUV will have a negative impact on the diagnosis of the model, but the conclusion obtained by separately extracting one health state from each working condition for testing is incomplete. The complete data set under three operating conditions shall be tested here.

6. The microrobots play an important role in the field of autonomous underwater vehicles. Please review and comment the recent work of microrobots in introduction((1) Frontiers in Bioengineering and Biotechnology, 2022, 10, 903219).

Round 2

Reviewer 1 Report

I have no comment to author.

My comments were fully reflected.

Reviewer 2 Report

The manuscript can be published in Sensors.